# The Influence of Frailty on Pharmacotherapy Adherence and Adverse Drug Reactions in Older Psychiatric Patients

**DOI:** 10.3390/geriatrics10020057

**Published:** 2025-04-07

**Authors:** Tuan Anh Phan, Rob Kok

**Affiliations:** 1Erasmus Medical Center, Hospital Pharmacy, Dr. Molewaterplein 40, 3015 GD Rotterdam, The Netherlands; 2Parnassia Group, Department of Old Age Psychiatry, Mangostraat 1, 2552 KS Den Haag, The Netherlands

**Keywords:** frailty, psychiatric patients, psychotropic drugs, treatment adherence, adverse drug reactions

## Abstract

**Aim/Objectives**: To investigate whether frailty predicts adherence to psychotropic drug treatment or adverse drug reactions, within 6 months after treatment initiation. **Methods**: A prospective cohort study including 77 patients over the age of 65, treated in one large psychiatric institute in the Netherlands. Patients were assessed at baseline for their frailty status, using different operationalizations of the Fried frailty criteria. Data on duration of psychotropic drug treatment and number of reported adverse drug reactions were retrieved from electronic patient files. Regression analyses were adjusted for age, sex, patient setting, and polypharmacy as potential confounders. **Results**: Frail patients were not significantly more likely to discontinue psychotropic treatment than non-frail patients (OR = 1.4; 95% CI 0.6–3.7, *p* = 0.468). Time to treatment discontinuation was also not statistically different between both study groups (HR = 0.8; 95% CI 0.4–1.6, *p* = 0.498), and neither was the number of adverse drug reactions (OR = 1.6, 95% CI 0.6–4.1, *p* = 0.345). **Conclusions**: We could not demonstrate a statistically significant effect of frailty as predictor of discontinuing psychotropic treatment or adverse drug reactions, but a lack of power may also explain our results. A more comprehensive frailty assessment may be needed to predict treatment adherence or adverse drug reactions in psychiatric patients.

## 1. Introduction

Frailty is defined as an increased susceptibility to adverse health outcomes and/or mortality when subjected to stressors [1]. Frailty is commonly operationalized according to Fried et al. by having at least three out of the following criteria: unintentional weight loss, weakness, exhaustion, slowness, and low physical activity [2]. The Fried criteria focus on the physical aspect of frailty, in contrast with the Frailty Index composed by a much longer and broader list of deficits in health that may include mood, cognition, sleep, and self-rated health [3]. Frailty predicts a series of adverse health outcomes, including adverse drug reactions, falls, hospitalization, disability, long-term care, and death [4]. Adverse drug reactions may also predispose to developing frailty in older people, and the relationship between frailty and adverse events may be bidirectional [5,6].

Only a few studies have analyzed whether frailty predicts adverse health outcomes in psychiatric patients. Frailty predicted admission to long-term care facilities, hospital admissions, and mortality in patients’ samples with a range of psychiatric disorders [7,8,9,10]. The majority of studies of older adults with psychiatric illnesses have been related to depression and have suggested a bi-directional association between frailty and depression [11,12]. A recent review of studies found only a few high-quality studies of the association between frailty and medication harm [13]. Psychotropic drugs with anticholinergic and sedative effects were a risk factor for developing frailty in community-dwelling older men [14], but we are not aware of studies that have analyzed whether frailty predicts adverse drug reactions to psychotropic drugs.

Another important adverse health outcome is medication nonadherence, as this may impede effective treatment of patients’ complaints and leads to poorer health outcomes and higher mortality [15]. Older patients are generally prescribed multiple drugs for physical diseases and are at increased risk of having adverse drug reactions. Several cohort studies have reported that frailty decreases medication adherence to somatic medications [16,17,18,19]. However, a prospective cohort study of chronic dialysis patients found frailty to be positively associated with medication adherence, perhaps because older patients were more concerned about their survival than about development of adverse drug effects [20].

Nonadherence to psychotropic drugs in the general population is very frequent. A meta-analysis found 56%, 50%, and 44% of patients to be nonadherent to their medication for schizophrenia, depression, and bipolar disorders, respectively [21]. Ineffective treatment or adverse drug reactions are the primary reasons for nonadherence to psychotropic drugs [21,22]. The risk of relapse or recurrence is considerably increased after discontinuation of antidepressants, antipsychotics, or mood stabilizers [23,24]. We are aware of only one study of the association between frailty and psychotropic nonadherence in psychiatric patients. In this study of 121 older outpatients treated with the antidepressants escitalopram or duloxetine, frail patients had a higher chance of switching antidepressants compared to non-frail patients, thus suggesting a higher nonadherence for frail patients [25]. Unfortunately, frailty is rarely taken into account in research regarding antidepressants for older adults [26]. If frailty predicted adverse drug reactions and treatment discontinuation in older patients, this would be of great importance in daily practice.

In the present study, we aimed to investigate the effect of frailty on nonadherence and adverse drug reactions in three main psychotropic drug groups, antidepressants, antipsychotics, and mood stabilizers, in older psychiatric patients. We analyzed whether frail psychiatric patients, compared to non-frail patients, (1) discontinued treatment with antidepressants, antipsychotics, and mood stabilizers more often or earlier within the first six months; and (2) experienced more adverse drug reactions within the first six months of psychotropic drugs treatment.

## 2. Materials and Methods

### 2.1. Study Design and Data Collection

This study is part of a larger ongoing longitudinal observational cohort study of frailty in older psychiatric patients at Parnassia Group, a large mental health institute in the Netherlands. Eligible participants for this study were inpatients and outpatients aged 65 years or older referred to the old age psychiatric clinic or the outpatient service. Patients who did not sufficiently master the Dutch language to answer our questionnaires, were diagnosed with a major neurocognitive disorder, or refused to participate were excluded. Patients were included from July 2015 until December 2021. All eligible patients were invited by research staff and included in the study after having given informed consent. Research staff used a structured assessment protocol and were trained in data collection including the frailty measures walking speed and handgrip strengths, in order to improve interrater reliability. As this study was not funded, research staff was not always available and then we were not able to recruit patients. Ethical considerations are described at the end of the manuscript.

### 2.2. Frailty Measurements

The baseline measurements took approximately 45 min on average and consisted of various validated questionnaires and two physical tests (hand grip strength and gait speed). The baseline measurements were assessed by trained research assistants within three weeks after intake or hospital admission. Measurements took place at the patient’s home in the case of ambulatory patients, and at our clinic in the case of inpatients. Frailty was assessed using the five Fried criteria: slowness, handgrip strength, exhaustion, unintentional weight loss, and low physical activity [2]. Slowness was measured using walking speed, which was calculated by measuring the time it took a patient to cover 4 m, with cutoff scores depending on gender and height based on the study by Fried et al. [2]. Handgrip strength was determined by taking the average of two measures using a Jamar handheld dynamometer and using a gender-specific cutoff value: a score of 30 kg for men and 18 kg for women, as determined by Fried et al. [2]. Exhaustion was defined as a score of 2 (frequently present) or 3 (usually or always present) on either of two questions (“I felt like everything I did was an effort” and “I couldn’t get ‘going’”) from the Center for Epidemiologic Studies Depression Scale (CES-D) during the previous week [27]. Weight loss was operationalized in the same way as in the NESDO study [28]. This criterion was met in the case of unintentional weight loss in the previous month, according to the Short Nutritional Assessment Questionnaire (SNAQ) [29]. Finally, the low physical activity criterion was met when the patient did not engage in any intense or moderately intense physical activity and less than 10 min per day walking, according to three questions from the self-administered version of the International Physical Activity Questionnaire (IPAQ) [30]. The reliability and validity of these rating scales are described in the references of each rating scale.

Frailty status was operationalized in four different ways: frailty as a dichotomous variable when patients met at least three frailty components (according to Fried), the number of frailty components, and the 2 physical components of frailty, grip strength, and gait speed.

### 2.3. Variable Definitions

Electronic Patient Files (EPF’s) were systematically reviewed to collect the following variables for our analyses: total number of drugs used, time to discontinuation (TTD) of psychotropic drugs, number of different psychotropic drug trials, and number of adverse drug reactions (ADR), all within 6 months after the start of their psychotropic drug use. Polypharmacy was defined as the use of 5 of more drugs (somatic and/or psychiatric) at baseline. Drugs prescribed for ‘as needed’ use were not counted. As these data are based on the prescription data of the hospital’s pharmacist, and nurses register whether the patient accepts or refuses medication; the validity of these data is expected to be high, except for adverse drug reactions (discussed below).

Only antidepressants, antipsychotics, and mood stabilizers initiated to treat the main psychiatric diagnosis (major depressive disorder, schizophrenia spectrum disorders, bipolar disorder, anxiety disorders, and other disorders) were recorded. Antipsychotics prescribed for a depressive disorder with melancholic features or for insomnia, antidepressants prescribed for neuropathic pain or insomnia, and benzodiazepines prescribed for any reason were excluded, as these medications are generally prescribed for a brief extent of time. For inpatients and outpatients, psychotropic drugs were prescribed under the supervision of old-age psychiatrists in specialized mental healthcare teams.

As successful drug treatment may lead to improvement in frailty measurements, we initially included only patients who started pharmacological treatment within 4 weeks after their baseline assessment (post-baseline group). However, roughly half of our patients started pharmacological treatment before the baseline frailty assessment, mainly because they needed (involuntary) treatment and could only be included in our study after their psychiatric condition improved enough to give informed consent for our study. We included these patients if the start of the psychotropic drug treatment was up to four weeks before the baseline frailty assessment (pre-baseline group).

Notes made by prescribing clinicians in the EPF’s were searched to verify why a drug was discontinued.

We only noted an ADR when it was confirmed by a clinician, regardless of the probability of its legitimately being an ADR and regardless of the claims of the patient. When an ADR caused by the same drug is documented on multiple occasions, it is recorded as a single ADR.

### 2.4. Statistical Analyses

All statistical analyses were performed with IBM SPSS Software version 27. To correct for multiple testing, the threshold for significance was set at *p* = 0.01 for all univariate and regression analyses. We first analyzed demographical and clinical characteristics to describe our study population. The normality of variables was tested with Shapiro–Wilk tests. As none of the variables were normally distributed (all *p* < 0.05), outcomes are presented as median with interquartile ranges for continuous variables. Categorical variables are presented as percentages.

We performed all statistical analyses with both the entire cohort (pre- and post-baseline groups) and solely the group of patients that started psychotropic treatment after the baseline frailty assessment: the post-baseline group. All the regression analyses were corrected for age, sex, patient setting, and polypharmacy as potential confounders.

For the first research question, univariate analyses of the influence of frailty on TTD within six months of treatment initiation were performed with a Chi-2 test, Mann–Whitney U tests, Kruskal–Wallis tests, and Pearson’s correlation tests with frailty as dichotomous variable and number of frailty components, grip strength, and gait speed as independent variables, respectively. TTD for the frail and non-frail group according to the Fried criteria was analyzed with Cox regression analyses with correction for all the potential confounders mentioned earlier. The assumption of proportional hazard of time for the Cox regression was met (all Cox with time-dependent variables, *p* > 0.05).

For the second research question, due to the low frequency of the number of patients with more than two documented ADR’s, we categorized the number of ADR’s as either 0, 1, or ≥2. Therefore, univariate analyses of the influence of frailty on the number of ADR’s were performed with Mann–Whitney U tests for dichotomous frailty as the independent variable, and Kruskal–Wallis tests for number of frailty components, with grip strength and gait speed as the independent variables. Furthermore, the influence of frailty on the number of ADR’s was analyzed with ordinal regression analyses corrected for the confounders mentioned earlier. The assumption of proportional odds was met with a test of parallel lines (all *p* > 0.05).

## 3. Results

We invited 185 eligible patients to participate in our study, of which 166 patients agreed. The remaining 19 patients were excluded because Covid-19 restrictions hindered physical contact (*n* = 10), because of refusal to participate (*n* = 6), or because they did not respond to invitation letters or phone calls (*n* = 3). The majority (n = 89) of the remaining 166 patients did not start with psychotropic drug treatment within one month of the baseline frailty assessment, resulting in a study group of 77 patients. Baseline demographic and clinical data are presented in Table 1. The median age of the cohort was 74, with the majority being female and inpatient. Frailty was present in 27 (35.1%) and polypharmacy in 43 (55.8%) of our patients. Two outliers were identified in grip strength and four in gait speed, defined as a deviation of more than three times the standard deviation from the mean. These values were omitted from the dataset.

We compared baseline characteristics between the pre-baseline group (*n* = 38) and post-baseline group (*n* = 39). The only significant difference between both groups was patient setting. Significantly more pre-baseline patients were inpatients compared to the post-baseline group (94.7% versus 66.7%; χ^2^(1) = 9.668, *p* = 0.003).

### 3.1. Time to Discontinuation

Frail patients were not significantly more likely to discontinue psychotropic treatment within 6 months than non-frail patients (44.4% versus 36%; OR = 1.4 with 95% CI 0.6–3.7, *p* = 0.468). No significant difference was observed in TTD between antidepressants, antipsychotics, and mood stabilizers within the entire cohort of 77 patients (Kruskal–Wallis H(2) = 0.861, *p* = 0.650) and within the post-baseline group (Kruskal–Wallis H(2) = 909, *p* = 0.635), and these psychotropic-drug classes were analyzed as one group.

We found no significant difference between frail and non-frail patients in univariate analyses of the time to discontinue treatment (HR = 0.8; 95% CI 0.4–1.6, *p* = 0.498). A Kaplan–Meier survival plot with both patient groups is presented in Figure 1. Within the post-baseline group, time until discontinuation also did not differ significantly between frail and non-frail patients (HR = 0.5; 95% CI 0.2–1.5).

A Kruskal–Wallis test conducted on the influence of number of frailty components on TTD did not show a significant effect for the entire cohort (H(5) = 2.894, *p* = 0.716), nor for the post-baseline group only (H(5) = 5.0101, *p* = 0.415). In similar vein, Pearson’s correlation tests presented a nonsignificant effect of grip strength (r(73) = −0.075, *p* = 0.521) and gait speed (r(69) = −0.010, *p* = 0.933) for the entire cohort. This applied to the post-baseline group as well, r(35) = 0.117, *p* = 0.489 and r(34) = −0.259, *p* = 0.128 respectively. The Cox regression analyses with TTD are presented in Table 2; none of the variables showed a statistically significant effect on TTD.

### 3.2. Number of Adverse Drug Reactions

The median number of adverse drug reactions confirmed by clinicians was 1 (IQR 0–2). The most prevalent adverse drug reactions were tremors (in 11 patients), orthostatic hypotension (in 7 patients), and dry mouth and dizziness (both in 5 patients). Univariate Mann–Whitney U tests did not show a significant difference in number of adverse drug reactions between frail and non-frail patients when conducted in the entire cohort (U = 560, *p* = 0.581) and in the post-baseline group (U = 161, *p* = 0.795). The OR for having adverse events also did not differ significantly between frail compared to non-frail patients (OR = 1.6, 95% CI 0.6–4.1, *p* = 0.345). Number of frailty components, grip strength, and gait speed also showed nonsignificant effects in univariate analyses on the number of adverse drug reactions as well (Table 3).

None of the frailty variables in the conducted ordinal regressions corrected for confounders yielded statistically significant results (Table 4). Although frail patients were approximately three times more likely to report ADRs than non-frail patients in the post-baseline group, this difference was not significant (OR = 2.8; 95% CI 0.58–13.6).

## 4. Discussion

Our study investigated the effect of frailty, operationalized with various indicators, on time to discontinue three psychotropic drug groups (antidepressants, antipsychotics, and mood stabilizers) and on number of adverse drug reactions in older psychiatric patients.

Almost 40% of our patients discontinued their treatment within six months in our study. However, frail patients did not differ significantly from non-frail patients in discontinuation rate nor in time to discontinue psychotropic drug treatment. The number of discontinuations of psychotropic drugs is in line with a meta-analysis of nonadherence to psychotropic drugs [21]. The nonsignificant effect of frailty on discontinuation in our study is in contrast with the only other study with psychotropic drugs of Brown et al., although that study only analyzed the effect on the number of patients who switched the antidepressants escitalopram or duloxetine [25]. Other differences between study designs and the higher number of included patients in the study by Brown et al. may explain these different results. The only finding in our study that may have clinical relevance is gait speed. With every additional second a patient takes in walking four meters, their chance of discontinuation increases almost 40% in the post-baseline group (23% in the entire cohort). The *p*-values of both findings were 0.049 and 0.047, respectively, and both were considered nonsignificant due to multiple testing.

In frail patients, clinicians reported 40% more often ADRs compared to non-frail patients, although this was also nonsignificant. The results nonetheless may still be clinically meaningful, and the low number of reported ADRs and the low number of patients may explain the non-significance. We cannot compare our results with other studies of ADRs of psychotropic drugs, as none of these included frailty as predictor. Our results are consistent with a higher burden of ADRs of somatic medication in frail patients treated compared to non-frail patients [13]. Polypharmacy also did not affect the number of ADR in our study. This is in contrast with previous findings for somatic medication, where polypharmacy generally results in a higher burden of ADRs, consequently prematurely discontinuing their treatment [14,19].

We would have expected a higher number of ADRs in our patient group with a median age of 74 years and a median of 5 somatic drugs. Almost all the patients also had physical diseases, and differentiating between physical complaints due to somatic diseases, somatic drug treatment, and psychotropic drug treatment may be very difficult. In daily practice, our severely depressed, psychotic, or manic patients are often not well motivated to accept psychotropic drugs, and treating physicians may be reluctant to attribute physical complaints as adverse drug reactions to their psychotropic drugs.

Differences in the results between the analyses performed in the entire cohort and the post-baseline group might be due to unjustified classification of some patients as frail. Frailty symptoms could have developed after starting medication, such as unintentional weight loss or weakness due to antidepressants. This distorts a possible causal relationship between frailty and our outcome variables, and therefore we presented all analyses with both the post-baseline group and with all patients.

Nevertheless, our findings deliver valuable insight into psychiatric patients’ drug treatment trajectories during drug treatment. Clinicians should be aware of the influence frailty has on these treatment trajectories and vice versa. From a practical standpoint, relatively simple questionnaires and physical assessments could greatly aid in anticipating potential adverse reactions to psychotropic drugs. Moreover, clinicians should focus on potential factors that may decrease nonadherence as described by Semahegn et al. [21]. With this knowledge, patients’ expectations can be managed accordingly. With proper information regarding their frailty status and drug treatment trajectory, their adherence to medication may be enhanced.

The strength of our study lies in the fact that it is a prospective study with typical patients in treatment in our wards, without the high attrition that is often found in randomized controlled trials. We used validated rating scales, and frailty assessments and data based on drug prescription are expected to have high validity. We employed multiple variables for indicators of frailty and could investigate the influence of frailty on both a dichotomous and a continuous scale. We included three major psychotropic drug classes, while the only prior study of psychotropic drugs focused on antidepressants only. Lastly, this is the first study to measure the impact of frailty on adverse drug reactions to psychotropic drug treatment.

This study has several limitations. The patient population was relatively small, and our nonsignificant results may be due to a lack of power. The low number of patients is partly due to lack of research staff during some longer periods, and partly due to the outbreak of the Covid pandemic shortly after we started including ambulatory patients. Second, our data on adverse drug reactions were largely dependent on existing information in EPF’s, which was not reported systematically. Although or study was prospective, we did not ask clinicians to register drug treatment details systematically between baseline and the follow-up assessment after one year. Therefore, the exact validity of these data is not known. For example, the well-known Naranjo scale, a standardized assessment of causality for all adverse drug reactions, was not used in daily practice by our clinicians [31]. Due to the retrospective assessment of ADRs in our study, it is not always clear if discontinuations were due to ADRs. Not all physicians’ reports clearly specified the rationale behind these discontinuations. Nonadherence could very well have had a multifactorial cause, such as a lack of efficacy or social support, or the presence of physical illnesses. However, we followed strict data extraction protocols to determine which data would be included in our study, which offsets the inconsistency in data provision. Our results may not be generalizable to other psychotropic drugs and to outpatients.

## 5. Conclusions

In summary, we found no statistical difference between frail and non-frail patients in discontinuing psychotropic treatment and in number of ADRs. However, this may be explained by a lack of power. Despite our negative results, we believe that frailty is still a potential predictor for anticipating adverse reactions to psychotropic drug treatment. We suggest that future studies should be prospective with larger datasets, using other operationalizations of frailty as, for example, the frailty index.

## Figures and Tables

**Figure 1 geriatrics-10-00057-f001:**
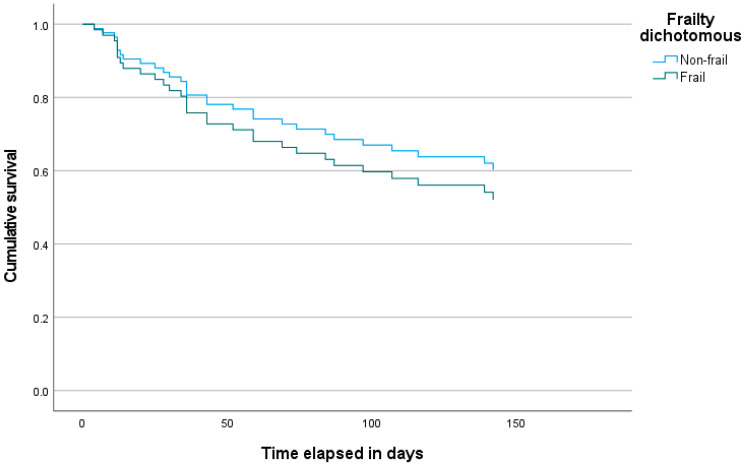
Kaplan–Meier plot with time to discontinue psychotropic drugs within 6 months (*n* = 77).

**Table 1 geriatrics-10-00057-t001:** Patient characteristics at baseline for the entire study group (*n* = 77).

Characteristics	
Age, mean (SD), range	73.9 (6.4), 65–93
Female sex, *n* (%)	56 (72.7)
Inpatient, *n* (%)	62 (80.5)
Psychotropic drug class	
Antidepressant	55
Antipsychotic	39
Mood stabilizer	25
Number of drugs, median (IQR), range	5 (3–7), 1–13
MMSE-score, median (IQR), range	27 (26–29), 19–30
Frail according to Fried criteria, *n* (%)	27 (35.1%)
Number of frailty components, median (IQR)	2 (1–3)
Grip strength in kilograms, median (IQR), range	47.5 (37.5–62.5), 14.5–104
Gait speed, median (IQR), range in seconds	4.5 (3.4–6.3), 1.8–9.4

Abbreviations: IQR = interquartile range; MMSE = Mini-Metal State Evaluation, *n* = number; SD = standard deviation.

**Table 2 geriatrics-10-00057-t002:** Cox regression analyses of the effect of dichotomous frailty, number of frailty components, grip strength, and gait speed on time-to-discontinuation psychotropic drugs in the entire cohort (*n*= 77) and the post-baseline group (*n* = 39).

Variable	Dichotomous Frailty			Number of Frailty Components			Grips Strength			Gait Speed		
	HR	95% CI	*p*	HR	95% CI	*p*	HR	95% CI	*p*	HR	95% CI	*p*
**Entire cohort**												
Frailty variable	1.08	0.48–2.43	0.148	1.12	0.84–1.49	0.457	1.0	0.98–1.04	0.513	1.23	1.0–1.50	0.049
Female sex	0.85	0.37–2.0	0.427	0.87	0.38–2.0	0.737	1.01	0.30–3.46	0.987	0.72	0.30–1.72	0.460
Age	1.05	0.99–1.11	0.029	1.05	0.99–1.11	0.091	1.06	0.99–1.12	0.052	1.04	0.98–1.11	0.203
Outpatient	0.75	0.28–2.0	0.505	0.79	0.29–2.12	0.683	0.82	0.31–2.18	0.693	0.93	0.33–2.58	0.886
Polypharmacy	1.02	0.89–1.17	0.074	1.00	0.87–1.16	0.998	1.04	0.90–1.19	0.620	0.99	0.85–1.15	0.868
**Post-baseline group**												
Frailty variable	2.05	0.71–5.94	0.188	1.40	0.97–2.02	0.070	1.00	0.97–1.03	0.792	1.39	1.00–1.92	0.047
Female sex	1.19	0.42–3.35	0.744	1.52	0.50–4.61	0.459	0.84	0.20–3.51	0.808	0.99	0.33–2.93	0.981
Age	1.07	0.98–1.1.6	0.137	1.07	0.97–117	0.102	1.07	0.98–1.17	0.133	1.02	0.93–1.12	0.739
Outpatient	0.59	0.19–1.82	0.355	0.60	0.19–1.91	0.385	0.55	0.16–1.90	0.346	0.81	0.25–2.66	0.734
Polypharmacy	0.98	0.81–1.18	0.839	0.95	0.78–1.16	0.609	1.00	0.83–1.22	0.979	0.95	0.77–1.17	0.594

Abbreviations: HR = hazard ratio.

**Table 3 geriatrics-10-00057-t003:** Kruskal–Wallis test results of the influence of number of frailty components, grip strength, and gait speed on the number of ADR’s in the entire cohort (*n* = 77) and the post-baseline group (*n* = 39).

Variable	Number of Frailty Components ^a^		Grip Strength ^b^		Gait Speed ^b^	
	H	*p*	H	*p*	H	*p*
Entire cohort	4.077	0.538	2.114	0.347	1.260	0.533
Post-baseline group	4.720	0.451	2.196	0.333	0.266	0.876

^a^ 5 degrees of freedom. ^b^ 2 degrees of freedom.

**Table 4 geriatrics-10-00057-t004:** Ordinal regression analyses results on the influence of dichotomous frailty, number of frailty components, grip strength, and gait speed and on the number of ADR in the entire cohort (*n* = 73) and the post-baseline group (*n* = 39).

Variable	Dichotomous Frailty			Number of Frailty Components			Grips Strength			Gait Speed		
	OR	95% CI	*p*	OR	95% CI	*p*	OR	95% CI	*p*	OR	95% CI	*p*
**Entire cohort**												
Frailty variable	1.34	0.52–3.49	0.544	1.12	0.80–1.57	0.503	0.99	0.96–1.02	0.509	1.18	0.91–1.51	0.207
Female sex	0.71	0.27–1.84	0.477	0.68	0.26–1.75	0.436	0.48	0.11–2.02	0.318	0.58	0.22–1.55	0.277
Age	1.27	0.96–1.10	0.503	1.02	0.95–1.09	0.651	1.03	0.96–1.10	0.435	1.01	0.94–1.09	0.791
Outpatient	0.62	0.19–2.13	0.431	0.65	0.20–2.13	0.481	0.66	0.20–2.12	0.499	0.69	0.21–2.28	0.538
Polypharmacy	0.71	0.83–1.13	0.674	0.96	0.82–1.13	0.601	0.97	0.83–1.14	0.724	0.94	0.79–1.11	0.438
**Post-baseline group**												
Frailty variable	1.41	0.35–5.75	0.631	1.14	0.71–1.82	0.595	1.00	0.95–1.04	0.814	1.18	0.79–1.75	0.417
Female sex	0.48	0.13–1.79	0.276	0.50	0.13–1.85	0.297	0.29	0.04–2.11	0.223	0.41	0.11–1.59	0.196
Age	0.99	0.90–1.11	0.957	1.00	0.90–1.11	0.960	1.00	0.90–1.11	0.979	0.98	0.87–1.10	0.713
Outpatient	0.41	0.09–1.82	0.239	0.41	0.09–1.86	0.246	0.39	0.08–1.94	0.247	0.41	0.09–1.90	0.254
Polypharmacy	0.95	0.75–1.18	0.618	0.94	0.75–1.18	0.577	0.93	0.73–1.18	0.554	0.89	0.70–1.14	0.361

Abbreviations: OR = odds ratio.

## Data Availability

Due to ethical reasons, the privacy-sensitive data that support the findings of this study are not publicly available.

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
