# Peer review of "The Influence of Frailty on Pharmacotherapy Adherence and Adverse Drug Reactions in Older Psychiatric Patients"

_geriatrics, 2025, doi:10.3390/geriatrics10020057_

Round 1
Reviewer 1 Report
Comments and Suggestions for Authors
A brief summary
The manuscript is a well-written valuable work. In addition, few studies with statistically non-significant results are published, so it is important that this study is published.
A total of 185 people were included in the study. What is the reason for the small number of patients, even though the study was long enough (line 81)?
I suggest that the minimum and maximum values of the continuous variables are shown in Table 1.
In general, the columns in Tables 2 and 4 are overlapping, it would be useful to separate them.
General concept comments
Article:
- Is the manuscript clear, relevant for the field and presented in a well-structured manner?
Yes
- Is the manuscript scientifically sound and is the experimental design appropriate to test the hypothesis?
yes
- Are the manuscript’s results reproducible based on the details given in the methods section?
Yes
- Are the figures/tables/images/schemes appropriate? Do they properly show the data? Are they easy to interpret and understand? Is the data interpreted appropriately and consistently throughout the manuscript? Please include details regarding the statistical analysis or data acquired from specific databases.
yes
- Are the conclusions consistent with the evidence and arguments presented?
- yes
- Please evaluate the ethics statements and data availability statements to ensure they are adequate.
Appropriate
Author Response
Comment 1. A total of 185 people were included in the study. What is the reason for the small number of patients, even though the study was long enough (line 81)?
Response 1: The low number of patients is partly due to lack of research staff during some longer periods, and the outbreak of the covid-pandemic shortly after we started including ambulatory patients. We have added this sentence in the limitation section.
Comment 2. I suggest that the minimum and maximum values of the continuous variables are shown in Table 1.
Response 2: we have added this in Table 1.
Comment 3. In general, the columns in Tables 2 and 4 are overlapping, it would be useful to separate them.
Response 3: we apologise for the difficult-to-read Tables, and added a grid to improve readability.
Reviewer 2 Report
Comments and Suggestions for Authors
In this study the authors asses the influence of fragility on medication adherence in older psychiatric patients.
It is interesting to study this group of patients, as is more usual to address younger people when evaluating adherence in the context of psychiatry.
There are some points that should be improved.
Material and Methods:
- I could not find in the text reference about the approval of the study by an Ethics Committee
- When recording polypharmacy it is not clear if all the drugs taken by the patient were considered or only those related with the psychiatric problem. Even though in the next paragraph the authors state that only these drugs were recorded. If this is the case, they should indicate that is a polypharmacy only referring to psychiatric treatment.
Results:
- Tables contain many values; they are somewhat difficult to read. Maybe it would help if the tables have a grid.
- What were the most frequent ADR?
It is readeable but could be improved.
Author Response
Comment 1. I could not find in the text reference about the approval of the study by an Ethics Committee.
Response 1. This information was at the end of the text, (see Institutional Review Board Statement and informed consent statement). However,we also added in the method section: All eligible patients were invited by trained research staff and included in the study after given informed consent.
Comment 2. When recording polypharmacy it is not clear if all the drugs taken by the patient were considered or only those related with the psychiatric problem. Even though in the next paragraph the authors state that only these drugs were recorded. If this is the case, they should indicate that is a polypharmacy only referring to psychiatric treatment.
Response 2. We apologise for not being more clear about this and added "(somatic and/or psychiatric)" in our sentence where we defined polypharmacy.
Comment 3. Tables contain many values; they are somewhat difficult to read. Maybe it would help if the tables have a grid.
Response 3. We fully agree and apologise, we added the grid to improve readability.
Comment 4. What were the most frequent ADR?
Response 4. We have added in paragraph 3.2.: The most prevalent adverse drug reactions were tremors (in 11 patients), orthostatic hypotension (in 7 patients) constipation, dry mouth and dizziness (both in 5 patients).
Reviewer 3 Report
Comments and Suggestions for Authors
Dear authors,
Thank you for the opportunity to review your interesting manuscript. I will give my feedback following the structure of the manuscript.
1.Title and abstract
The title is informative, and the abstract effectively summarizes the manuscript's key aspects. However, the aim of the study appears below the abstract title. I suggest adding 'Aim/Objective' after the abstract title for clarity.
2.Background
I would like to congratulate the authors on this chapter. In my view, it provides valuable information and is well-written and well-organized. However, it might be interesting to explore how these patients are prescribed psychotropic drugs in the Netherlands. Is this treatment more closely linked to hospital care or primary care? Understanding how different healthcare settings function could add valuable context for the reader.
3.Material and Methods
I have some suggestions for the authors that could help to improve this section:
Study Design and Data Collection
This section is clear, but it would be interesting to know how the participants were invited to take part in the study. Who invited them, and where did this take place? The authors mention that this study is part of a larger ongoing longitudinal study, but were the participants aware that their data would be collected for this purpose? In this regard, it would be valuable to include a specific section on ethical considerations, covering aspects such as ethics committee approval and informed consent
Frailty measurements
This section is clear. I only suggest adding information about where the measurements took place (e.g., primary care, home, etc.) and including references for the first tests such as Fried criteria…(if they have been previously validated).
Variable definitions
This section is clear. Nothing to add.
Statistical analysis
This section is very clear. Nothing to add.
4.Results
I would like to congratulate the authors for the results obtained. They are very clear and well presented. Nothing to add.
5.Discussion and Conclusions.
The section is comprehensive and discusses important aspects of the study results. It also presents the strengths and limitations. Additionally, the conclusions are clear.Nothing to add
I hope the review helps improve your work.
Best regards,
Author Response
Comment 1. However, the aim of the study appears below the abstract title. I suggest adding 'Aim/Objective' after the abstract title for clarity.
Response 1. We have added in the abstract Aim/Objective.
Comment 2. However, it might be interesting to explore how these patients are prescribed psychotropic drugs in the Netherlands. Is this treatment more closely linked to hospital care or primary care? Understanding how different healthcare settings function could add valuable context for the reader.
Response 2. We have added in the paragraph Variable Definition: "For inpatients and outpatients, psychotropic drugs were prescribed under the supervision of old-age psychiatrists in specialized mental health care teams."
Comment 3. it would be interesting to know how the participants were invited to take part in the study. Who invited them, and where did this take place? The authors mention that this study is part of a larger ongoing longitudinal study, but were the participants aware that their data would be collected for this purpose? In this regard, it would be valuable to include a specific section on ethical considerations, covering aspects such as ethics committee approval and informed consent.
Response 3. we apologise for not being more clear about this and added: "All eligible patients were invited by trained research staff and included in the study after given informed consent. " We also added "Measurements took place at people’s home for ambulatory patients, and at our clinic for inpatients." Ethical considerations were at the end of our manuscript, in the Institutional Review Board Statement and the Informed Consent Statement, but we agree that is makes sense to present the informed consent procedure also in the text.
Comment 4. I only suggest adding information about where the measurements took place (e.g., primary care, home, etc.) and including references for the first tests such as Fried criteria…(if they have been previously validated).
Response 4. See previous response for the first remark, and we added the references for the Fried criteria in the section of frailty measurements.
Reviewer 4 Report
Comments and Suggestions for Authors
Authors investigated whether frailty is associated with adherence to psychotropic drug treatment or adverse drug reactions. A prospective cohort study included 77 patients over the age of 65. Patients were assessed at baseline for their frailty status, Data on duration of psychotropic drug treatment and number of reported adverse drug reactions were retrieved from electronic patient files.
I see some issues with statistical analyses:
- When presenting results of regression analysis in tables, authors should show Ors and Hrs with 95% CI and p values. All other values (ß, SE) are not really relevant and do not help the interpretation.
- When authors conducted Cox Regression, they for sure can show also results of Kaplan Meier curves.
- Lines 161-162 “Furthermore, the influence of frailty on the number of ADR’s was analyzed with ordinal regression analyses corrected for the confounders” What do ordinal regression analyses mean? Do they mean logistic regression analyses?
- Abstract; authors write “Frail patients were not significantly more likely to discontinue psychotropic treatment than non-frail patients (χ2(1) = 0.526, p = 15.468). Time to treatment discontinuation was also not statistically different between both study groups (Mann-Whitney U = 624, p = .582), neither was the number of adverse drug reactions (Mann-Whitney U = 560, p = .581).” I recommend to show HR and OR with 95% CI rather than U tests which are univariable and do not enable a good interpretation of results.
- Please add study country to Abstract
Author Response
Comment 1. When presenting results of regression analysis in tables, authors should show Ors and Hrs with 95% CI and p values. All other values (ß, SE) are not really relevant and do not help the interpretation.
Response 1. We fully agree and thank the reviewer for this comment. we have changed this in both the tables and the text.
Comment 2. When authors conducted Cox Regression, they for sure can show also results of Kaplan Meier curves.
Response 2. We have added this as Figure 1.
Comment 3. Lines 161-162 “Furthermore, the influence of frailty on the number of ADR’s was analyzed with ordinal regression analyses corrected for the confounders” What do ordinal regression analyses mean? Do they mean logistic regression analyses?
Response 3. Logistic regression can only be done if a variable has 2 outcomes, like having or not having adverse events. Ordinal regression is a regression analyses for an ordinal variable that has more than 2 values, in our case the groups with either 0, 1, of 2 or more adverse events. This has more power than logistic regression. We hope the reveiuwer can agree with our choice.
Comment 4. Abstract; authors write “Frail patients were not significantly more likely to discontinue psychotropic treatment than non-frail patients (χ2(1) = 0.526, p = 15.468). Time to treatment discontinuation was also not statistically different between both study groups (Mann-Whitney U = 624, p = .582), neither was the number of adverse drug reactions (Mann-Whitney U = 560, p = .581).” I recommend to show HR and OR with 95% CI rather than U tests which are univariable and do not enable a good interpretation of results.
Response. This in line with the excellent first comment and we have changed this also in the abstract.
Comment 5. Please add study country to Abstract.
Response 5. we have added in the abstract: "....in The Netherlands."
Round 2
Reviewer 4 Report
Comments and Suggestions for Authors
-
Author Response
Thank you very much for the many suggestions to improve our manuscript.
We added the motivation in the introduction, lines 67-68.
We mentioned that this is a longitudinal observational study in line 79 but added 'cohort' in order to be more clear about this important point. We also added the STROBE checklist as attachment.
We added reliability and validity of the rating scales in line 113-114, in lines 126-129 about the outcome variables, and also added more information about these topics in the strengths and limitations section (lines 361-363, 375)
We have trained all research personal to avoid information bias and added this in lines 86-89.
We added information about recruitment in lines 89-90.
We only registered the number of eligible patients, refusals and drop-out when research staff was available.This Information is presented in lines 183-186.
We described the pre/post baseline groups in lines 138-145. We created confusion by calling the pre-baseline group the 'entire cohort' in the tables, and added an explanation in lines 161-162.
We described the ethical consideration in lines 402-406 but we added this in lines 90-91. We will add more information int the text is that would be better according to your judgement.
We added suggestions for future studies in the conclusion (lines 390-392).
We added the specification concerning the statistical data analyses (line 397).